# Quality of Life in Older Patients after a Heart Failure Hospitalization: Results from the SENECOR Study

**DOI:** 10.3390/jcm11113035

**Published:** 2022-05-27

**Authors:** Daniele Luiso, Marta Herrero-Torrus, Neus Badosa, Cristina Roqueta, Sonia Ruiz-Bustillo, Laia C. Belarte-Tornero, Sandra Valdivielso-Moré, Ronald O. Morales, Olga Vázquez, Núria Farré

**Affiliations:** 1Heart Failure Unit, Cardiology Department, Hospital del Mar, 08003 Barcelona, Spain; d.luiso@gmail.com (D.L.); nbadosa@psmar.cat (N.B.); sruiz@psmar.cat (S.R.-B.); lbelarte@psmar.cat (L.C.B.-T.); svaldivielso@psmar.cat (S.V.-M.); romoralesmurillo@psmar.cat (R.O.M.); 2Department of Medicine, Universitat Autónoma de Barcelona, 08193 Barcelona, Spain; croqueta@psmar.cat; 3Geriatrics Department, Hospital del Mar, 08003 Barcelona, Spain; mherrero@psmar.cat (M.H.-T.); ovazquez@psmar.cat (O.V.); 4Biomedical Research Group on Heart Disease, Hospital del Mar Medical Research Group (IMIM), 08003 Barcelona, Spain; 5Department of Medicine, Universidad Pompeu Fabra, 08002 Barcelona, Spain

**Keywords:** quality of life, heart failure, older patients, prognosis

## Abstract

Background: Information about health-related quality of life (HRQoL) in heart failure (HF) in older adults is scarce. Methods: We aimed to describe the HRQoL of the SENECOR study cohort, a single-center, randomized trial comparing the effects of multidisciplinary intervention by a geriatrician and a cardiologist (intervention group) to that of a cardiologist alone (control group) in older patients with a recent HF hospitalization. Results: HRQoL was assessed by the short version of the disease-specific Kansas Cardiomyopathy Questionnaire (KCCQ-12) in 141 patients at baseline and was impaired (KCCQ-12 < 75) in almost half of the cohort. Women comprised 50% of the population, the mean age was 82.2 years, and two-thirds of patients had preserved ejection fraction. Comorbidities were highly prevalent. Patients with impaired HRQoL had a worse NYHA functional class, a lower NT-proBNP, a lower Barthel index, and a higher Clinical Frailty Scale. One-year all-cause mortality was 22.7%, significantly lower in the group with good-to-excellent HRQoL (14.5% vs. 30.6%; hazard ratio 0.28; 95% confidence interval 0.10–0.78; *p* = 0.014). In the group with better HRQoL, all-cause hospitalization was lower, and there was a trend towards lower HF hospitalization. Conclusions: The KCCQ-12 questionnaire can provide inexpensive prognostic information even in older patients with HF. (Funded by grant Primitivo de la Vega, Fundación MAPFRE. ClinicalTrials number, NCT03555318).

## 1. Introduction

Heart failure (HF) is one of Western society’s major public health problems. The epidemiological dimension of HF, its clinical complexity, the impact on patients’ quality of life, and the burden it represents for a health system with finite resources [1] make this syndrome one of the greatest health, organizational, and economic challenges of the present day.

The clinical practice guidelines of the European Society of Cardiology [2] establish that the main goals of the treatment of patients with HF are to improve quality of life, reduce mortality, and reduce hospitalizations. Classically, the efficacy endpoint used to evaluate new therapies in HF is to reduce mortality. On the one hand, mortality has the advantage that it is a strong and an easy-to-measure event. On the other hand, it has an important disadvantage: being the final manifestation of the disease, it does not represent the clinical course until the fatal outcome, or the evolution of those patients who do not die [3]. Thus, considering that HF is a chronic and progressive disease with florid symptoms and significant repercussions on functionalism, an ideal efficacy endpoint should reflect both the symptoms and the patient’s subjective perception of their health status [3,4,5,6]. In this way, assessing health-related quality of life (HRQoL) as an efficacy endpoint in HF is crucial. It provides precious information on both the patients who survive and those who die. It has been shown that HRQoL in HF correlates well with both disease severity and mortality and allows cost-effectiveness evaluations when implementing new therapeutic options [7,8]. The measurement of HRQoL is easy and inexpensive since it is carried out through questionnaires that can be generic or specific to the disease. HRQoL is a multidimensional concept that includes four fundamental aspects: physical, psychological, social, and functional status. The multidimensional nature of HRQoL allows for capturing a complete perspective of the patient. The impairment of HRQoL in HF is reflected, above all, in the functional dimension, with particular repercussions in the domains that inform about mobility and activities of daily living [9].

Information on HRQoL in HF in older adults is scarce. Most of the data reported in the literature on HRQoL in HF come from studies that include non-older patients and patients with reduced left ventricular ejection fraction (LVEF) [8,10,11,12,13,14,15]. Describing HRQoL and its correlation with prognosis in older people with HF could provide valuable clinical information, since the improvement in HRQoL in this population could have an even higher value than a reduction in mortality, both for patients and health professionals [16].

## 2. Materials and Methods

### 2.1. Study Design

The SENECOR study was a single-center, randomized trial comparing the effects of multidisciplinary intervention by a geriatrician and a cardiologist (intervention group) to that of a cardiologist alone (control group) in older patients with a recent HF hospitalization. The primary endpoint for the trial was all-cause hospitalization. Quality-of-life assessment was a pre-specified secondary endpoint of the SENECOR study. The Ethics Committee approved the study (number 2017/7653/I) and all patients signed written informed consent forms. The details of the study design and results have been published [17] and the trial is registered with ClinicalTrials.gov (NCT03555318). Briefly, patients 75 years or older and hospitalized due to HF were randomized to a follow-up performed by a cardiologist (usual care) or by a cardiologist and a geriatrician. All patients were assessed with the Canadian Study of Health and Aging (CSHA) Clinical Frailty Scale during hospitalization [18]. Frailty was defined as a CSHA equal to or higher than 4. Functional status was assessed with the Lawton [19] and Barthel index [20] and cognitive status with the Spanish version of the Pfeiffer Questionnaire (Short Portable Mental Status Questionnaire (SPMSQ)) [21]. The 12-item Kansas City Cardiomyopathy Questionnaire (KCCQ-12) was used to assess HRQoL specifically related to HF [22]. The functional class was evaluated by the New York Heart Association (NYHA) classification. In patients randomized to the intervention group, the geriatrician assessed the social sphere with the Gijón socio-family assessment scale (abbreviated and modified) (Barcelona version) [23], the emotional sphere with the Geriatric Depression Scale Short Form (GDS-SF) Yesavage [24], nutritional status with the Mini Nutritional Assessment Short Form (MNA-SF) [25] and plasma albumin, and the presence of geriatric syndromes. After the geriatrician assessment and depending on the patient’s needs, up to eighteen interdisciplinary interventions were carried out in each area evaluated. The study showed that the multidisciplinary intervention by the cardiologist and geriatrician was associated with a decrease in all-cause hospitalization at one-year follow-up (62.7% in the intervention group and 77.3% in the control group) (hazard ratio 0.67; 95% confidence interval 0.46–0.99; *p* = 0.046) [17].

In the SENECOR study, the calculated sample size to detect a statistically significant difference between the two groups was 114 patients in the intervention group and 114 patients in the control group for 1 year [17]. However, patients with exclusion criteria or who refused to participate were higher than expected, and the estimated patient goal was not reached. On the other hand, the number of events was much higher than anticipated. Of the 150 patients who were finally included in the SENECOR study, we only included in the present study patients who had answered the KCCQ-12 at baseline, leaving a sample size of 141 patients (Figure 1).

### 2.2. Quality-of-Life Assessment

The KCCQ-12 is the short version (12-item) of the Kansas City Cardiomyopathy Questionnaire (KCCQ) (23-item). This self-administered test measures symptoms, physical and social limitations, and quality of life in patients with HF. It has been validated in HF both with reduced and preserved ejection fractions [8,26]. Moreover, it has proven to be both reproducible and sensitive to important changes in HF health status [26,27,28,29]. The shorter version has shown to be more feasible to implement while preserving the psychometric properties of the full instrument [22]. Scores for each domain are summarized by the KCCQ summary score, which has values between 0 and 100, with higher scores indicating better HF-specific health status. An increase of fewer than 5 points is considered a small clinical change [28]. Several studies have established a KCCQ-12 cut-off point of 75 or higher to identify patients with good-to-excellent HRQoL [30]. Therefore, we considered HRQoL impaired if KCCQ-12 was below 75. In the SENECOR study, KCCQ-12 was measured during the baseline visit. At one-year follow-up, all the baseline assessments including KCCQ-12 were repeated in those who survived.

### 2.3. Study Outcome

The main aim was to evaluate whether a good-to-excellent HRQoL was associated with lower all-cause mortality at one-year follow-up.

The secondary objectives were to evaluate whether a good-to-excellent HRQoL was associated with lower all-cause hospitalization and HF hospitalization at one-year follow-up and evaluate the extent of change in the KCCQ-12 scores at one-year follow-up.

### 2.4. Statistical Analysis

Mean and standard deviation were used to describe continuous variables, and numbers and proportions to describe the categorical variables. The chi-square test or Fisher’s exact test for categorical variables and the Student’s t-test for continuous variables were used to assess the baseline differences between patients with KCCQ-12 below and over 75.

Time-to-event data were evaluated using Kaplan–Meier estimates and Cox proportional-hazards models. The adjusted hazard ratio (HR) of HF hospitalization for HRQoL measured by the KCCQ-12 was analyzed using Cox proportional hazard models. The models were adjusted for potential confounders selected among patient characteristics that were significantly associated with a better HRQoL status. We included all variables with *p* < 0.05. We decided to include age and gender due to their known prognostic value.

Study data were collected and managed using REDCap electronic data capture tools hosted at Parc de Salut Mar [31,32]. REDCap (Research Electronic Data Capture) is a secure, web-based software platform designed to support data capture for research studies, providing (1) an intuitive interface for validated data capture; (2) audit trails for tracking data manipulation and export procedures; (3) automated export procedures for seamless data downloads to common statistical packages; and (4) procedures for data integration and interoperability with external sources.

## 3. Results

One hundred and fifty patients were randomized between 2 July 2018 and 15 November 2019. A total of 141 patients answered the KCCQ-12 at baseline and were included in the analysis. Figure 1 shows the flow diagram of the study.

HRQoL was impaired in almost half of the cohort. Only 2 patients (1.4%) had very-poor-to-poor HRQoL (KCCQ-12 0–24), 30 patients (21.3%) had poor-to-fair HRQoL (KCCQ-12 25–49), and 40 patients (28.4%) a fair-to-good HRQoL (KCCQ-12 50–74). A good-to-excellent HRQoL (KCCQ 75–100) was present in 48.9% of patients at the baseline visit. Women comprised 50% of the population, the mean age was 82.2 years, and two-thirds of patients had HF with a preserved ejection fraction. Comorbidities were highly prevalent. Baseline characteristics were not different between patients with impaired and non-impaired HRQoL (Table 1).

The only statistically significant differences were a lower NYHA functional class and a surprisingly higher NT-proBNP and left ventricular mass index in the group with better HRQoL. These patients also had a higher Barthel index and a lower Clinical Frailty Scale (Table 2).

One-year all-cause mortality was 22.7% and was significantly lower in the group with good HRQoL (14.5% vs. 30.6%; hazard ratio 0.28; 95% confidence interval 0.10–0.78; *p* = 0.014). In the group with better HRQoL, all-cause hospitalization was lower, and there was a trend towards lower HF hospitalization (Figure 2) (Table 3).

In patients with a KCCQ-12 measured at one year, there was a statistically significant increase in KCCQ-12. KCCQ-12 went from 71.5 ± 21.5 to 83.1 ± 20.8, *p* < 0.001, and 69.6% of patients had good-to-excellent HRQoL (Figure 3).

Of the 72 patients with impaired HRQoL at baseline, 44 repeated the KCCQ-12 assessment at 12 months. Of those, 15 patients did not improve HRQoL (KCCQ-12 < 75) and 29 patients improved (KCCQ-12 > 75). Baseline characteristics of patients with impaired HRQoL at baseline who did not improve and who improved to a good-to-excellent HRQoL at one-year follow-up are summarized in Table 4 and Table 5. A lower proportion of all-cause hospitalization was found in the group with HRQoL improvement, although statistical significance was not reached (62.1% vs. 86.7%, *p* = 0.09).

## 4. Discussion

In our study involving older patients with a recent HF hospitalization, almost half of the patients had impaired HRQoL measured by the KCCQ-12 questionnaire. Surprisingly, baseline characteristics did not allow the identification of patients with worse HRQoL, except for parameters usually associated with aging, such as frailty and functional status measured by Barthel index, NYHA functional class, NT-proBNP, and left ventricular mass index. A good-to-excellent HRQoL was significantly associated with lower one-year all-cause mortality and hospitalization. In patients with HRQoL measured at one year, there was a significant improvement in the KCCQ-12 score.

This study extends prior works describing the association between HRQoL and clinical outcomes. It has already been shown that KCCQ provides prognostic information independent of other clinical data in patients with HF [8,10,11,12,13,14,15]. However, none of these studies examined the prognostic significance of KCCQ-12 in a prospective elderly cohort with a recent admission for HF. Indeed, our series differs from those previously published in two relevant aspects that should be noted. First, with a mean age of 82.2 years, our population was more than 10 years older than the oldest cohort published to date [33]. Second, the patients’ profiles were rather different from what has been published so far. In fact, the prevalence of HF with a preserved ejection fraction was higher than in previous studies (66.7%), probably concerning the age of the population. Finally, the presence of comorbidities was substantial.

It has been reported that, among a cohort of stable patients with HF, no significant changes were detected by the KCCQ questionnaire at mid-term follow-up. In contrast, large changes were observed among a cohort of patients recovering from admission for decompensated HF [27]. Since the HRQoL assessment in the SENECOR study had a median (interquartile range) of 6 (5–9) after discharge from decompensated HF, our results are in line with previous evidence. Interestingly, patients with impaired HRQoL at baseline who did not have HRQoL improvement at one-year follow-up were more likely to be women. This is consistent with the previous finding that, in a cohort of patients with HF and reduced ejection fraction, women reported significantly worse HRQoL than men, although HRQoL was independently associated with outcome similarly in men and women [34]. On the other hand, patients who did not improve HRQoL also had a higher baseline LVEF than patients who improved. This could be explained by a potential improvement in LVEF over time in the group with a lower baseline LVEF, which could be associated with improvements in HRQoL. However, it could also reflect the several pitfalls that the actual classification of HF based on LVEF values has [35]. Moreover, in our study, a trend towards a lower proportion of all-cause hospitalization was found in the group with HRQoL improvement. Anyway, we must consider a possible selection bias in this analysis due to patients who died or did not repeat the HRQoL assessment at a one-year follow-up.

Better strategies are needed to help physicians efficiently target healthcare resources to HF patients at the highest risk. Our findings suggest that noninvasive risk stratification based on HRQoL measurement by the KCCQ-12 questionnaire can provide prognostic information even in older patients with HF, which could be an essential reference for subsequent treatment decisions when identifying candidates for disease management for whom increased care may reduce hospitalization and prevent death. Future studies are needed to establish whether the assessment of HRQoL in older HF patients with questionnaires such as KCCQ-12 can improve outcomes. It is worth noting that the baseline characteristics did not allow us to identify patients with worse HRQoL. Hence, HRQoL should be assessed in all patients to identify high-risk patients.

### Limitations

Since this was a single-center study with a relatively small sample size, our data must be interpreted with caution. Moreover, HRQoL measurements in our study were administered as a part of routine outpatient follow-up visits within a clinical trial. In the setting of a clinical trial, the self-perception of HRQoL may increase regardless of the intervention due to multiple factors (extra care, more intensive management, optimism, etc.) [36,37]. Whether HRQoL assessments will have similar prognostic value outside this setting remains to be established. Finally, although our results were adjusted for multiple demographic and clinical patient factors, a possibility of residual unmeasured confounding factors cannot be definitively excluded, and our findings need to be validated in a larger-cohort multicenter study.

## 5. Conclusions

In older patients with a recent hospital admission for HF, good-to-excellent HRQoL was significantly associated with lower one-year all-cause mortality and hospitalization. These data support the assessment of HRQoL in relation to HF in the older population.

## Figures and Tables

**Figure 1 jcm-11-03035-f001:**
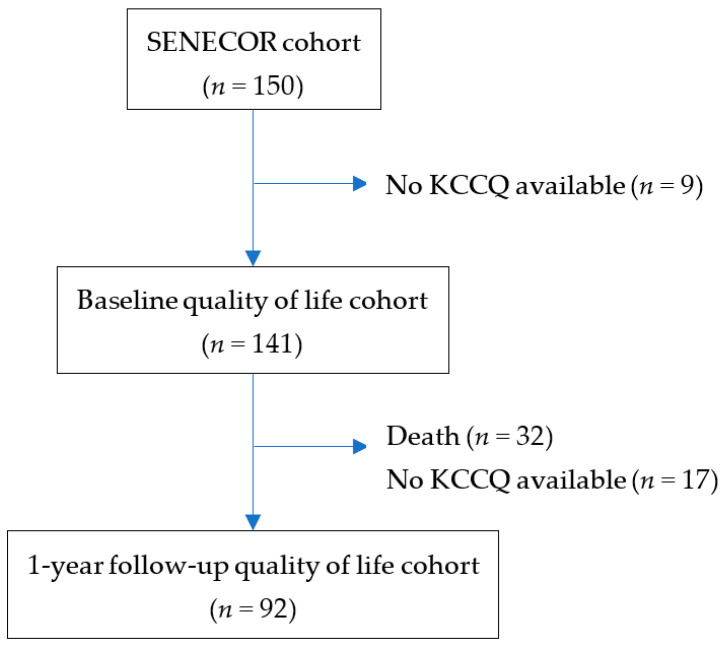
Flow diagram of the study.

**Figure 2 jcm-11-03035-f002:**
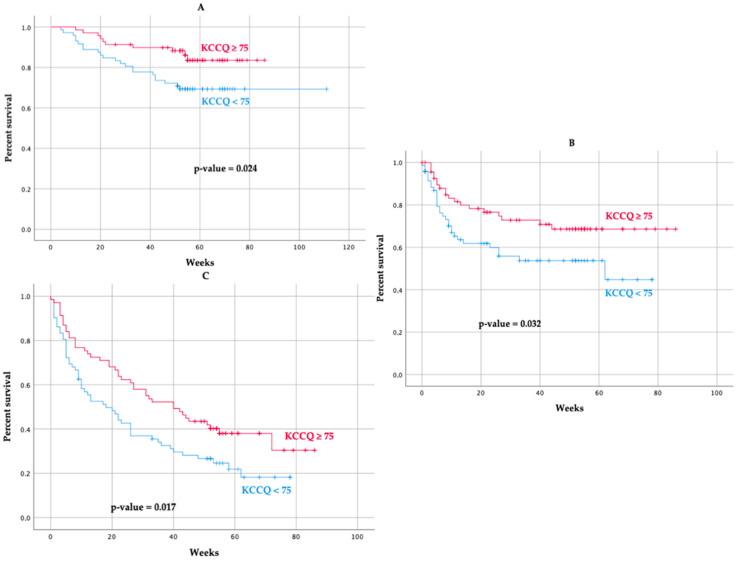
Unadjusted Kaplan–Meier for (**A**) all-cause death, (**B**) heart failure hospitalization, and (**C**) all-cause hospitalization.

**Figure 3 jcm-11-03035-f003:**
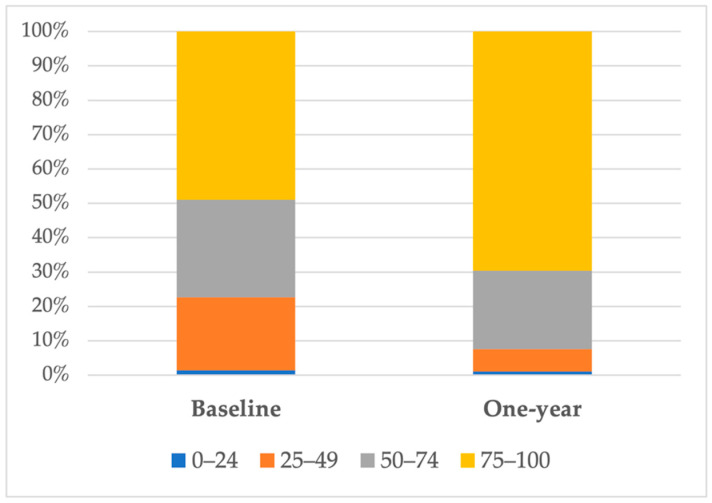
Change in KCCQ-12 at one-year follow-up.

**Table 1 jcm-11-03035-t001:** Baseline clinical and demographic characteristics of the patients included in the study.

	KCCQ < 75 (*n* = 72)	KCCQ 75–100 (*n* = 69)	*p*-Value
Age (years)	81.7 ± 4.8	82.3 ± 4.7	0.43
Female	37 (51.4)	34 (49.3)	0.80
Hypertension	63 (90)	62 (89.9)	0.98
Diabetes mellitus	31 (44.3)	28 (41.2)	0.71
Dyslipidemia	47 (66.2)	41 (59.4)	0.41
Stroke/TIA	9 (13.4)	10 (15.4)	0.75
Chronic kidney disease	54 (75)	44 (63.8)	0.15
Anemia	42 (58.3)	39 (56.5)	0.83
Peripheral vascular disease	9 (12.7)	14 (20.6)	0.21
Chronic lung disease	28 (38.9)	18 (26.1)	0.11
Cancer	16 (22.5)	19 (27.9)	0.43
Myocardial infarction	18 (25)	11 (15.9)	0.18
Coronary percutaneous intervention	14 (19.4)	10 (14.5)	0.43
TAVI or Mitraclip	1 (1.4)	2 (2.9)	0.48
Cardiac surgery:			
CABG	2 (2.8)	3 (4.3)	
Valve replacement	4 (5.6)	6 (8.7)	0.60
CABG and valve replacement	3 (4.2)	2 (2.8)	
Atrial fibrillation or flutter	54 (75)	43 (62.3)	0.10
Moderate-to-severe valve disease	22 (31.4)	22 (32.8)	0.86
Device therapy:			
Pacemaker	12 (16.7)	10 (14.5)	0.52
CRT or ICD	1 (1.4)	4 (5.7)	
Previous history of HF	43 (59.7)	38 (55.1)	0.58
Duration of HF *:			
<3 months	12 (27.9)	3 (7.9)	
3–6 months	1 (2.3)	3 (7.9)	
6–12 months	4 (9.3)	5 (13.2)	0.18
1–5 years	17 (39.5)	15 (39.5)	
>5 years	9 (20.9)	11 (28.9)	
HF hospitalization the previous year *	19 (45.2)	12 (32.4)	0.25
HF categories:			
HFpEF (LVEF ≥ 50%)	48 (66.7)	46 (66.7)	
HFmrEF (LVEF 40–49%)	9 (12.5)	6 (8.7)	0.70
HFrEF (LVEF < 40%)	15 (20.8)	17 (24.6)	
Ecocardiographic parameters:			
LVEF (%)	52.1 ± 13.6	52.7 ± 15.2	0.79
Left ventricular mass index (g/m^2^), *n* = 134	120.2 ± 30.7	134.2 ± 36.7	0.018
TAPSE (mm), *n* = 126	17.5 ± 4.3	17.3 ± 3.6	0.72
Right ventricle (mm), *n* = 88	28.9 ± 6.7	29.7 ± 7.3	0.56
Heart failure etiology			
Ischaemic	10 (14.1)	12 (17.4)	
Hypertensive	11 (15.5)	12 (17.4)	
Dilated cardiomyopathy	4 (5.6)	6 (8.7)	0.16
Valve heart disease	21 (29.6)	17 (24.6)	
Other/unknown	25 (35.2)	22 (31.9)	
Medications at discharge:			
ACEI/ARB-II/ARNI	35 (49.3)	39 (57.4)	0.34
MRA	9 (12.7)	12 (17.6)	0.41
Betablockers	52 (73.2)	49 (72.1)	0.87
Diuretics	68 (95.8)	67 (98.5)	0.62
Anticoagulation	53 (74.6)	46 (67.6)	0.36
Antiplatelet therapy	12 (16.9)	14 (20.6)	0.58
Oral antidiabetic drugs	24 (34.3)	20 (29.4)	0.54
Insulin	14 (19.7)	10 (14.7)	0.43
Proton-pump inhibitors	48 (67.6)	46 (67.6)	1.00
Statin	50 (70.4)	37 (54.4)	0.051
Calcium channel antagonists	25 (36.2)	17 (25.0)	0.15
Nitrates	16 (22.5)	10 (14.7)	0.24
Hydralazine	10 (14.1)	7 (10.3)	0.50
Amiodarone	16 (22.9)	8 (11.8)	0.09
Digoxin	3 (4.3)	1 (1.5)	0.62
Vitamin D supplements	25 (35.2)	20 (29.4)	0.47
Oral iron supplements	19 (26.8)	19 (27.9)	0.88
Benzodiazepines	16 (22.5)	14 (20.6)	0.78
Antidepressant drugs	20 (28.2)	16 (23.5)	0.53
Bronchodilators	27 (38.0)	20 (29.4)	0.28

Data are numbers (percentage) or mean ± standard deviation. ACEI: angiotensin-converting enzyme inhibitors; ARB-II: angiotensin II receptor blockers; ARNI: angiotensin receptor and neprilysin inhibition; CABG: coronary artery bypass grafting; CRT: cardiac resynchronization therapy; HF: heart failure; HFrEF: heart failure with reduced ejection fraction. HFmrEF: heart failure with mildly reduced ejection fraction; HFpEF: heart failure with preserved ejection fraction; ICD: implantable cardioverter defibrillator; LVEF: left ventricular ejection fraction; MRA: mineralocorticoid receptor antagonists; TIA: transient ischemic attack; TAPSE: tricuspid annular plane systolic excursion; TAVI: transcatheter aortic valve implantation. * Only for patients with a previous history of HF.

**Table 2 jcm-11-03035-t002:** Hospitalization and first appointment characteristics.

	KCCQ < 75 (*n* = 72)	KCCQ 75–100 (*n* = 69)	*p*-Value
NT-proBNP at discharge, pg/mL	1977.5 (950.5–3917.0)	2774.5 (1767.0–6191.5)	0.018
High-sensitivity T troponin (Hs-TnT) at discharge, ng/L	37.5 (26.6–65.1)	43.7 (30.5–70.2)	0.26
eGFR (mL/min) at discharge	46.4 ± 19.9	47.3 ± 20.4	0.81
Frailty (Clinical Frailty Scale) ≥ 4	44 (61.1)	27 (40.3)	0.014
Clinical Frailty Scale	4.2 ± 1.4	3.7 ± 1.1	0.02
Barthel index	81.8 ± 19.7	90.4 ± 12.3	0.002
Basic activities of daily living (Barthel index):			
Independent (100)	17 (23.6)	25 (36.2)	
Minimally dependent (61–99)	45 (62.5)	41 (59.4)	0.07
Partially to totally dependent (0–60)	10 (13.9)	3 (4.3)	
Instrumental activities of daily living (Lawton index)	4.6 ± 2.3	5.3 ± 1.9	0.054
Pfeiffer Short Portable Mental Status Questionnaire (SPMSQ)	1 (1–3)	1 (0–2)	0.08
NYHA functional class	2.5 ± 0.6	2 ± 0.4	<0.001
Intervention geriatrician and cardiologist	31 (43.1)	40 (58)	0.08
KCCQ-12 at baseline	53 ± 15.9	88.3 ± 7.8	<0.001

Data are number (percentage), mean ± standard deviation, or median (interquartile range). eGFR: estimated glomerular filtration rate; NT-proBNP: N-terminal prohormone of brain natriuretic peptide; NYHA: New York Heart Association; KCCQ-12: Kansas City Cardiomyopathy Questionnaire-12.

**Table 3 jcm-11-03035-t003:** Primary and secondary outcomes during follow-up.

	KCCQ < 75 (*n* = 72)	KCCQ 75–100 (*n* = 69)	*p*-Value
All-cause mortality	22 (30.6)	10 (14.5)	0.014
All-cause hospitalization	55 (76.4)	43 (62.3)	0.017
HF hospitalization	30 (41.7)	19 (27.5)	0.051

Data are numbers (percentage). HF: heart failure. The model is adjusted for age, female sex, Barthel index, Clinical Frailty Scale, NT-proBNP value at discharge, New York Heart Association functional class, and Kansas City Cardiomyopathy Questionnaire 75–100.

**Table 4 jcm-11-03035-t004:** Baseline clinical and demographic characteristics of the patients included in the study according to the improvement of KCCQ-12 at one-year follow-up.

	No KCCQ Improvement (*n* = 15)	KCCQ Improvement (*n* = 29)	*p*-Value
Age (years)	80.0 ± 4.4	81.9 ± 5.08	0.23
Female	11 (73.3)	12 (41.4)	0.04
Hypertension	14 (93.3)	25 (89.3)	1.00
Diabetes mellitus	8 (53.3)	14 (50)	0.84
Dyslipidemia	11 (73.3)	19 (65.5)	0.74
Stroke/TIA	2 (13.3)	4 (14.8)	1.00
Chronic kidney disease	11 (73.3)	22 (75.9)	1.00
Anemia	9 (60)	14 (48.3)	0.46
Peripheral vascular disease	2 (14.3)	4 (13.8)	1.00
Chronic lung disease	6 (40)	9 (31)	0.55
Cancer	2 (13.3)	10 (34.5)	0.17
Myocardial infarction	2 (13.3)	4 (13.8)	1.00
Coronary percutaneous intervention	2 (13.3)	7 (24.1)	0.69
TAVI or Mitraclip	1 (7.1)	0 (0)	0.33
Cardiac surgery:	2 (13.3)	4 (13.8)	0.69
Atrial fibrillation or flutter	10 (66.7)	23 (79.3)	0.47
Moderate to severe valve disease	4 (28.6)	8 (27.6)	1.00
Device therapy:			
Pacemaker	12 (16.7)	10 (14.5)	0.52
CRT or ICD	1 (1.4)	4 (5.7)	
Previous history of HF	7 (46.7)	18 (62.1)	0.33
HF hospitalization the previous year *	2 (28.6)	5 (29.4)	1.00
LVEF (%)	62.3 ± 3.9	49.3 ± 13.2	<0.001

Data are number (percentage) or mean ± standard deviation. HF: heart failure; LVEF: left ventricular ejection fraction; TIA: transient ischemic attack; TAVI: transcatheter aortic valve implantation. * Only for patients with a previous history of HF.

**Table 5 jcm-11-03035-t005:** Hospitalization and first appointment characteristics according to the improvement of KCCQ-12 at one-year follow-up.

	No KCCQ Improvement (*n* = 15)	KCCQ Improvement (*n* = 29)	*p*-Value
NT-proBNP at discharge, pg/mL	1162 (606.6–3579.0)	1799.5 (801.9–3562.5)	0.44
High-sensitivity T troponin (Hs-TnT) at discharge, ng/L	31.6 (22.9–44.5)	46.9 (22.4–73.0)	0.21
eGFR (mL/min) at discharge	52.9 ± 23.2	46.0 ± 22.1	0.36
Frailty (Clinical Frailty Scale) ≥ 4	12 (80.0)	16 (55.2)	0.11
Clinical Frailty Scale	4.5 ± 1.4	3.9 ± 1.1	0.14
Barthel index	76.2 ± 22.0	86.1 ± 15.3	0.09
Instrumental activities of daily living (Lawton index)	4.5 ± 2.3	4.6 ± 2.0	0.82
Pfeiffer Short Portable Mental Status Questionnaire (SPMSQ)	1 (1–2)	1 (1–3)	0.62
NYHA functional class	2.7 ± 0.7	2.2 ± 0.6	0.053
Intervention geriatrician and cardiologist	8 (53.3)	15 (51.7)	0.92
KCCQ-12 at baseline	55.1 ± 15.3	52.6 ± 17.5	0.64

Data are number (percentage), mean ± standard deviation, or median (interquartile range). eGFR: estimated glomerular filtration rate; NT-proBNP: N-terminal prohormone of brain natriuretic peptide; NYHA: New York Heart Association; KCCQ-12: Kansas City Cardiomyopathy Questionnaire-12.

## Data Availability

The data presented in this study are available on request from the corresponding author. The data are not publicly available due to ethical restrictions.

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
