# Peer review of "Quality of Life in Older Patients after a Heart Failure Hospitalization: Results from the SENECOR Study"

_jcm, 2022, doi:10.3390/jcm11113035_

Round 1
Reviewer 1 Report
The manuscript by Daniele Luiso et al. entitled “Quality of life in older patients after a heart failure hospitalization. Results from the SENECOR study” aimed to describe HRQoL of the SENECOR study cohort, a single-center, randomized trial comparing the effects of a multidisciplinary intervention by a geriatrician and a cardiologist (intervention group) to that of a cardiologist alone (control group) in older patients with a recent HF hospitalization.
The article is quite well written, however, this is a single-center trial with a sample size of only 141 patients. Moreover, a possibility of residual unmeasured confounding cannot be definitively excluded. It would be important to report the different classes of drugs taken by patients and echocardiographic parameters of right heart function, such as TAPSE.
Author Response
Point 1: The article is quite well written, however, this is a single-center trial with a sample size of only 141 patients. Moreover, a possibility of residual unmeasured confounding cannot be definitively excluded.
Response 1: Thank you very much for your kind comment. We agree with the reviewer. Although our results were adjusted for multiple demographic and clinical patient factors, the possibility of residual unmeasured confounding cannot be definitively excluded, and our findings need to be validated in a larger-cohort multicenter study. We have added this limitation to the discussion.
Point 2: It would be important to report the different classes of drugs taken by patients and echocardiographic parameters of right heart function, such as TAPSE.
Response 2: Thank you very much for this suggestion. We have added in Table 1 information about echocardiographic parameters (left ventricular mass index, TAPSE, right ventricle diameter) and medication at discharge.
Reviewer 2 Report
Please refer to manuscript for the comment and suggestion.

Author Response
Point 1: all highlighted should be comma (,) and d should be capital D.
Response 1: We have modified it as per the reviewer's suggestion.
Point 2: add on briefly about this KCCQ in introduction sentence.
Response 2: Thank you for this comment. We have added in the abstract that the KCCQ-12 is a disease-specific HRQoL questionnaire. The Materials and Methods section provides a detailed description of the KCCQ-12.
Point 3: is to reduce mortality / an easy...
Response 3: We have modified it as per the reviewer's suggestion.
Point 4: “The SENECOR study was a single-center, randomized trial comparing the effects of a multidisciplinary intervention by a geriatrician and a cardiologist (intervention group) to that of a cardiologist alone (control group) in older patients with a recent HF hospitalization.” state the duration and year of study done.
Response 4: Thank you for this suggestion. The study was carried out between 2 July 2018 and 15 November 2019. We have added the information about the duration and the year of the SENECOR study in the Results section.
Point 5: “The KCCQ-12 is the short version (12-item) of the Kansas City Cardiomyopathy Questionnaire (KCCQ) (23-item).” why need to shorten the questionaire. pls provide the questionaire in supplemantary.
Response 5: The shorter version of the KCCQ has shown to be more feasible to implement while preserving the psychometric properties of the full instrument (doi:10.1161/CIRCOUTCOMES.115.001958). After consulting with cvoutcomes, the questionnaire is copyright protected, and we do not have permission to add it as a Supplementary material.
Point 6: “In the SENECOR study, KCCQ-12 was measured during the baseline visit and at a one-year follow-up.” please state here what is done to this patient here. even it is concluded in flow. what happen to patient that died within one year follow up?
Response 6: In the SENECOR study, KCCQ-12 was measured during the baseline visit. At one-year follow-up, all the baseline assessments, including KCCQ-12 were repeated in those who survived. We have clarified the manuscript in the Materials and Methods section.
Point 7: “SENECOR cohort (n = 150)” how do u get this number? is it based on sample size?
Response 7: In the SENECOR study, was calculated that 114 patients in the intervention group and 114 patients in the control group would be necessary to detect as statistically significant the difference between the two groups. Therefore, taking into account the number of admissions for HF in our Cardiology Department, it was considered that it would be feasible to complete the inclusion of patients in 1 year. However, patients with exclusion criteria or who refused to participate were higher than expected, and therefore the estimated patient goal has not been reached, and the sample size was only 150 (intervention = 75, control = 75). On the other hand, the number of events was much higher than expected so the main aim was reached. Whe have added this explanation in the Materials and Methods section.
Point 8: Figure 2. label Y axis. can't see
Response 8: We have modified it as per the reviewer's suggestion.
Point 9: “No KCCQ improvement (n = 15). KCCQ improvement (n = 29).” how do u get this n number can you added this in the flow.
Response 9: Thank you for pointing this out. Of the 72 patients with impaired HRQoL at baseline, 44 repeated the KCCQ-12 assessment at 12 months. Of those, 15 patients did not improve HRQoL (KCCQ-12 < 75) and 29 patients improved (KCCQ-12 > 75). We have clarified the manuscript in the Results section.
Point 10: “Interestingly, patients with impaired HRQoL at baseline who did not have HRQoL improvement at one-year follow-up were more likely to be women with higher left ventricular ejection fraction.” eloborate why.
Response 10: The reviewer raises a very interesting point. It has already been reported that amongst patients with HFrEF, women reported significantly worse QoL than men. QoL was independently associated with subsequent outcomes, similarly in men and women. The KCCQ in general, and the KCCQ-OS in particular, showed the strongest independent association with outcome (doi:10.1002/ejhf.2154). We have added this reference to the Discussion.
Reviewer 3 Report
The article is interesting and it focuses on an often forgotten argument in patients with HF: the quality of life. It is important to define life quality, mostly in older patients and patients with advanced HF, in order to decide the best treatment strategy according to HF severity.
The work is well done and well written. Some aspects should be improved:
1) please add information regarding therapy of patients in Tables. May therapy have an impact on life quality ?
2) Actual classification regarding HF show several pitfalls. For this reason is mandatory an improvement of classification systems regarding HF patients (see J Clin Med. 2022 Feb 6;11(3):857. doi: 10.3390/jcm11030857 and J Card Fail. 2017 Apr;23(4):280-285. doi: 10.1016/j.cardfail.2016.12.002), in order to address patients to best treatment strategy such as palliative care or more intensive up tritation.
3) Please define also the etiology of HF
Author Response
Point 1: The article is interesting and it focuses on an often forgotten argument in patients with HF: the quality of life. It is important to define life quality, mostly in older patients and patients with advanced HF, in order to decide the best treatment strategy according to HF severity. The work is well done and well written.
Response 1: Thank you very much for your kind comment.
Point 2: 1) please add information regarding therapy of patients in Tables. May therapy have an impact on life quality ?
Response 2: Thank you very much for this suggestion. We have added in Table 1 information about medication at baseline, and we have not found statistically significant differences between patients with KCCQ-12 <75 and KCCQ-12 75-100.
Point 3: 2) Actual classification regarding HF show several pitfalls. For this reason is mandatory an improvement of classification systems regarding HF patients (see J Clin Med. 2022 Feb 6;11(3):857. doi: 10.3390/jcm11030857 and J Card Fail. 2017 Apr;23(4):280-285. doi: 10.1016/j.cardfail.2016.12.002), in order to address patients to best treatment strategy such as palliative care or more intensive up tritation.
Response 3: The reviewer raises a very interesting point. In fact, in our study, patients with impaired HRQoL at baseline who did not improve HRQoL at one-year follow-up had a higher LVEF than patients who did improve. This result is not consistent with previous research, which supports that the actual classification of HF based on LVEF values has several pitfalls and is mandatory an improvement of classification systems regarding HF to address patients to best treatment strategy. We have added the suggested references to the discussion.
Point 4: 3) Please define also the etiology of HF
Response 4: We have added the etiology of HF.
Round 2
Reviewer 1 Report
The manuscript by Daniele Luiso et al. entitled “Quality of life in older patients after a heart failure hospitalization. Results from the SENECOR study” aimed to describe HRQoL of the SENECOR study cohort, a single-center, randomized trial comparing the effects of a multidisciplinary intervention by a geriatrician and a cardiologist (intervention group) to that of a cardiologist alone (control group) in older patients with a recent HF hospitalization.
However, a possibility of residual unmeasured confounding cannot be definitively excluded.